# Synthesis and Electrochemical Performance of V_6_O_13_ Nanosheets Film Cathodes for LIBs

**DOI:** 10.3390/ma15238574

**Published:** 2022-12-01

**Authors:** Fei Li, Haiyan Xu, Fanglin Liu, Dongcai Li, Aiguo Wang, Daosheng Sun

**Affiliations:** 1Anhui Key Laboratory of Advanced Building Materials, Anhui Jianzhu University, Hefei 230022, China; 2Key Laboratory of Functional Molecule Design and Interface Process, Anhui Jianzhu University, Hefei 230601, China

**Keywords:** V_6_O_13_, film electrodes, nanosheet, phase transition

## Abstract

V_6_O_13_ thin films were deposited on indium-doped tin oxide (ITO) conductive glass by a concise low-temperature liquid-phase deposition method and through heat treatment. The obtained films were directly used as electrodes without adding any other media. The results indicate that the film annealed at 400 °C exhibited an excellent cycling performance, which remained at 82.7% of capacity after 100 cycles. The film annealed at 400 °C with diffusion coefficients of 6.08 × 10^−12^ cm^2^·s^−1^ (Li^+^ insertion) and 5.46 × 10^−12^ cm^2^·s^−1^ (Li^+^ extraction) in the V_6_O_13_ film electrode. The high diffusion coefficients could be ascribed to the porous morphology composed of ultrathin nanosheets. Moreover, the film endured phase transitions during electrochemical cycling, the V_6_O_13_ partially transformed to Li_0.6_V_1.67_O_3.67_, Li_3_VO_4_, and VO_2_ with the insertion of Li^+^ into the lattice, and Li_0.6_V_1.67_O_3.67_, Li_3_VO_4_, and VO_2_ partially reversibly transformed backwards to V_6_O_13_ with the extraction of Li^+^ from the lattice. The phase transition can be attributed to the unique structure and morphology with enough active sites and ions diffusion channels during cycles. Such findings reveal a bright idea to prepare high-performance cathode materials for LIBs.

## 1. Introduction

The increasing requirement for energy storage devices and novel clean energy resources has generated unparalleled research interest with the development of society [1,2,3,4]. In the recent few decades, lithium-ion batteries (LIBs) have been extensively applied in portable electronic devices and electric vehicles due to high energy density, power density, and long cyclic performance [5,6,7,8,9]. The electrode materials dominate the performance of LIBs [10]. Therefore, developing optimized electrode materials is key to improving the properties of LIBs. Layered vanadate oxides such as V_6_O_13_, V_2_O_5_, V_3_O_7_, VO_2_(B), and V_3_O_8_ have been considered as a sort of applicable material because of the multiple ions oxidation states, high capacity, and layered crystal structure for ions intercalation [11,12,13,14].

Mixed-valence V_6_O_13_ has been considered as a potential candidate for LIBs, which is a desired material compared with other vanadium oxides [15,16]. According to reports, it has a high theoretical capacity of 417 mA h g^−1^, corresponding to 8 Li per formula unit (vanadium being reduced from an average charge of +4.33 to +3), and a theoretical energy density of 890 W h kg^−1^ [17,18,19,20]. The excellent performance of V_6_O_13_ is attributed to its open structure, which is built up of single and double V-O layers with vanadium ions in multiple valence states (V^4+^ and V^5+^) [11,21]. Herein, the unique structure provides more ions diffusion paths and sufficient active sites, resulting in a high specific capacity and good cyclic performance.

However, there still exists plenty of possibilities for improvement in the performance of V_6_O_13_. Some achievements such as morphological regulation, structural adjustment, and so forth have been generally proposed as approaches, since the morphology and structure remarkedly affect the electrochemical properties. The morphology and structure of materials are decided on by synthesis methods. So far, various methods such as hydrothermal synthesis, sol-gel, electrodeposition, and thermal decomposition have been applied to prepare V_6_O_13_ [16,22]. Among these, the solution-based method has become a preferred choice for many scholars, owing to its unique advantages of low cost, conciseness, suitability for extensive application, and low temperature. The materials possessing a high surface area, small pores, stable structure, and nanometer size are more competitive, which promote electrolyte penetration and a shorter ion diffusion path and prevent self-aggregation. Zhong et al. synthesized hollow microflowers assembled from nanosheets with a large surface area, which exhibited a relatively high specific capacity (326 mA h g^−1^ at 0.1 A g^−1^), a great capacity retention, and a long cycle life when used as a cathode material for LIBs [18]. Fei et al. also fabricated V_6_O_13_ microflowers via the hydrothermal method to obtain high performance cathodes [16]. Wu et al. synthesized 2D V_6_O_13_ nanosheets with good crystallinity and robust configuration [23], Zou et al. also successfully synthesized V_6_O_13_ nanosheets that exhibited a high discharge capacity [24]. Moreover, there are some other interesting morphologies, including nanobelts, nanorods and so on, basically with a relatively large surface area [7,25]. Therefore, the influence of the morphology and structure on the electrochemical properties are of concern.

In the paper, V_6_O_13_ film electrodes were prepared via a low-temperature liquid phase deposition method and subsequent heat treatment under N_2_ atmosphere. Compared with other methods, the low-temperature liquid phase deposition is more simple, inexpensive, gentle, and the film prepared is uniform. Film electrodes provide better contact with the electrolyte and can be applied directly in LIBs without additional media. Furthermore, the film electrodes can be easily taken out during cycling, which facilitates the study of electrochemical mechanisms. It can be intuitively observed that any changes of the films occur during the cycles, such as color, integrity, and so forth. 

In conclusion, the advantages can be summarized in a few aspects: (1) Convenient preparation of thin film electrodes. The V_6_O_13_ films are directly applied to LIBs without adding additives, which facilitates and clarifies the electrochemical process; (2) simplicity and gentleness of the experimental method. The films are synthesized under mild conditions in a water bath without high temperature or pressure. (3) The excellent morphology and integrity of crystalline. The nanosheet morphology provides sufficient active sites and channels for ion diffusion. Moreover, the selected method ensures that the film has a high degree of crystallinity. Therefore, the V_6_O_13_ film electrodes are expected to obtain a breakthrough in the application of LIBs.

## 2. Experimental Section

### 2.1. Preparation of Films

All experimental drugs and reagents used in this experiment were analytically pure and could be used directly without further purification. 

The V_6_O_13_ films on ITO conductive glass were prepared via a low-temperature liquid-phase deposition method and heat treatment under N_2_ atmosphere. A deposition solution was prepared by dissolving 0.1630 g vanadium sulfate and 0.0840 g lactic acid in 50 mL deionized water. The deposition solution was magnetically stirred for 2 h in air at room temperature. After stirring, the pH was adjusted to 4.0 with ammonia. 

A piece of ITO conductive glass (2 × 2 cm^2^, electrical resistance of 6 Ω/□) was ultrasonically cleaned in detergent and deionized water for several times. After that, the washed ITO substrate was put face down horizontally into the deposition solution in the beaker, which was placed in a water bath at 90 °C for 3 days. When the reaction was completed, the substrate was taken out, rinsed slowly with deionized water, cooled naturally to room temperature, and dried to obtain the desired deposited precursor films. The precursor films were annealed under nitrogen atmosphere with a velocity of flow of 50 mL/min at 350 °C, 400 °C, and 450 °C at a rate of 2 °C/min for 2 h. The annealed films were obtained and named N−350, N−400, and N−450 to represent the films at the corresponding annealing temperature.

### 2.2. Materials Characterizations

The X-ray diffraction (XRD) patterns of the films were characterized by X-ray diffraction instrumentation (Rigaku Smart Lab 9kw, Tokyo, Japan) using Cu Kα radiation (λ = 1.5406 Å). Raman spectrum (Renishaw, England) was collected to obtain structural information. The further structural information of annealed films was measured by infrared spectroscopy (Nicolet 6700, Waltham, MA, USA). The microscopic morphology of the sample surface was examined with a scanning electron microscope (JEOL JSM7500F, Akishima, Japan). The chemical bonding states were characterized by X-ray photoelectron spectroscopy (Thermo ESCALAB 250Xi, Waltham, MA, USA) measurement with Al Kα source. To avoid the effects of oxide layers for the film exposed to air, etching was carried out prior to XPS testing.

### 2.3. Electrochemical Characterizations

The properties of films were tested in a three-electrode system with a CHI-604E electrochemical station (CH Instruments Inc., Shanghai, China) using 1M LiClO_4_/PC as the electrolyte. The prepared films were used directly as the working electrode and the effective area was 0.6 cm^2^, the platinum plate was used as the counter electrode, and the Ag/AgCl electrode in KCl saturated solution was used as the reference electrode. The three-electrode system was filled with nitrogen flow for 20 min to remove oxygen. Cyclic voltammetry (CV) was performed to investigate the electrochemical properties of the thin films. CV data were collected between −1 V and 0.4 V at different scanning rates of 1–10 mV·s^−1^.

## 3. Results and Discussion

### 3.1. Characterization of the Obtained Films

The XRD patterns of the crystal phase structure at different annealing temperatures are shown in Figure 1. The deposited film possesses a strong peak at 9.3° attributed to the hydrated phase of (NH_4_)_8_(V_19_O_41_(OH)_9_)(H_2_O)_11_ (JCPDS No. 78-2016). After annealing at 350 °C, the N−350 in Figure 1b displays two new peaks at 9.2° and 25.5° of NH_4_V_4_O_10_ (JCPDS No. 31-0075), indicating the gradual dehydration with temperature increases. After annealing at 400 °C and 450 °C, the peaks of NH_4_V_4_O_10_ disappeared completely, and the peaks at 15.1° and 25.3°, 26.8°, 33.5°, and 45.5° are observed in Figure 1b,c, which can be well indexed to monoclinic V_6_O_13_ phase (JCPDS No. 71-2235) with the lattice parameters of a = 11.922 Å, b = 3.68 Å, c = 10.138 Å, and β = 100.87°. The results manifest that V_6_O_13_ films have been formed from the precursor (NH_4_)_8_(V_19_O_41_(OH)_9_)(H_2_O)_11_ films through heat treatment.

The FTIR spectra of the structural information are shown in Figure 2. The FTIR spectra of the films are dominated by strong absorption in the 1020–525 cm^−1^ region and these absorption peaks are associated with the vibrations of the bonds between V and O [3]. The bands between 1000–900 cm^−1^ can be attributed to V=O stretching vibrations [3,26]. The band at 1406 cm^−1^ is the bending pattern of N-H vibrations [27,28], which only presents in the N−350 film, representing that NH_4_^+^ fully disappears when the annealing temperature increases, which is consistent with the above XRD results. The absorption peak is produced at around 875 cm^−1^ pointing to V-O stretching vibrations (shorter V-O bonds) [29]. The band at 714 cm^−1^ is a V-O-V stretching vibration due to bridging oxygen bonds (longer V-O bonds) [28,30]. There are no other absorption peaks, indicating almost no other impurities in the films. These results are consistent with the XRD results.

The Raman spectra of the films are displayed in Figure 3. The band at 993 cm^−1^ represents the V-O stretching vibration, while the one at 687 cm^−1^ can be ascribed to the V-O-V stretching vibration [31,32]. The peak at 523 cm^−1^ is attributed to the triple coordination oxygen (V_3_-O) stretching mode [30]. The peaks at 406 cm^−1^ and 281 cm^−1^ are due to the bending vibration of the V=O bond [33,34]. The additional low-frequency peaks at 139 and 189 cm^−1^ correspond to the stretching pattern of (V_2_O_2_)_n_ [35]. These peaks are in agreement with the results of previous investigations. The vibrational absorption peaks are observed at similar locations of the films, indicating that the compositions are basically the same. The Raman results are constant with the IR results.

Figure 4 shows the SEM images of the annealed films. The samples show an interesting flower-like morphology, in which the nanosheets stack to form a porous flower-like cluster microstructure. The nanosheets in N−400 have a length of about 1 μm and a width of about 8 nm, and the edges of the nanosheets appear at irregular steps left by the crystalline growth. With increasing the annealing temperature to 450 °C, the nanosheets (Figure 4c,d) grow thicker (~20 nm thick), and the step corners of the V_6_O_13_ nanosheets become rounder. The porous flower-like morphology of N−400 and N−450 possesses a large specific surface area, which is conducive to electrolyte penetration and ion migration. The thickness of the nanosheets increases with the annealing temperature, which may also affect the electrochemical properties caused by different diffusion paths.

### 3.2. Electrochemical Performance Study

CV curves of the film electrodes at different scan rates within the range of −1 to 0.4 V are shown in Figure 5. The CV curves of the N−400 and N−450 show a pair of obvious redox peaks and two pairs of weak redox peaks, indicating a multi-step insertion/extraction of the Li^+^ in V_6_O_13_ electrode. For the N−400, the reduction peaks are observed at about −0.38, −0.67, and −0.82 V, corresponding to the insertion of Li^+^ into the V_6_O_13_ electrode ($V_{6}O_{13}+{\mathrm{xLi}}^{+}+{\mathrm{xe}}^{-}={\mathrm{Li}}_{X}V_{6}O_{13}$). The oxidation peaks at about −0.21 V, −0.12 V, and −0.55 V correspond to the extraction of Li^+^ (${\mathrm{Li}}_{x}V_{6}O_{13}-{\mathrm{yLi}}^{+}-{\mathrm{ye}}^{-}={\mathrm{Li}}_{x-y}V_{6}O_{13}$). For the N-450, the oxidation peaks are located at −0.16 V, −0.06 V, and −0.44 V, and the reduction peaks are located at −0.40, 0.63, and 0.80 V, respectively. A tiny new reduction peak at around −0.61 V becomes remarkable with the decrease of the scan rate, which may be caused by a more adequate response due to the reduced scan rate. Comparing the CV curves of the N−400 and N−450, the current of the N−400 is larger than N−450. This can be attributed to the morphology and crystallinity determined by the annealing temperature, the crystallinity annealed at 450 °C becomes complete, and the lithium ions are difficult to insert/extract. In addition, a slight shift of peak position is observed with the increasing scan rate, which is mainly due to the polarization effect [36,37]. From the CV curves at different scan rates, the diffusion coefficient of Li^+^ in the V_6_O_13_ film electrode would be calculated from the peak current (*I_p_*) and scan rate.

Figure 6 represents the fitted straight lines of the peak anode and peak cathode currents, respectively, versus the open square *v*^1/2^ for different sweep rates. The Li^+^ diffusion coefficient in the film electrode is calculated according to the Randles–Sevcik equation.
(1)$$I_{p}=2.69\times 10^{5}An^{3/2}D^{1/2}Cv^{1/2}$$

In this equation, *I_p_* is the peak current (A); *A* is the effective area of the prepared V_6_O_13_ film electrode (cm^2^); *n* is the number of electrons transferred in the redox reaction during the process (here, for lithium batteries, the fixed value is 1); *D* is the diffusion coefficient of lithium ions (cm^2^ s^−1^); *C* is the number of electrons in the active ion of the electrolyte (the concentration of lithium ions in the electrolyte) (mol/cm^3^); and *v* is the scan rate (V/s). From Figure 6 can be seen that the *I*_p_ varies linearly with *v*^1/2^, the linear relationship indicates that the reaction is a diffusion-controlled process and the current in the redox process is limited by the diffusion of ions of the electrode surface. The calculated diffusion coefficients of Li^+^ in the N−400 and N−450 are listed below in Table 1. The diffusion coefficients of Li^+^ are same as other literatures [3,38,39]. The Li^+^ diffusion coefficient of N−400 is larger than N−450 for both the anode and cathode, suggesting that the diffusion of Li^+^ of the N−400 is easier than the N−450, which is due to the surface effect of the thinner nanosheets. Moreover, the diffusion coefficient of the cathode is significantly higher than anode, which means it is easier to embed lithium ions than to remove them and this result may lead to lithium ion retention in the interlayer. Hence, the lithium ions may partly be retained in the interlayer to cause capacity fading. The retention of lithium ions may be one of the reasons for the polarization of the CV curves.

The cycling performance of films is examined at a scan rate of 10 mV/s from −1 to 0.4 V in Figure 7. The CV curves of the N−400 similarly display one obvious pair and two in-distinctive pairs of redox peaks, which is consistent with the results in Figure 5. The integral area of CV curves can represent capacity to some extent ($C=\left(\int idV\right)/2vV$). For the N−400, the CV curves for the first 50 cycles are largely overlapping, with little capacity fading, the peak area starts to fade slowly for the next 50 cycles. The integral area of the CV curves maintains 83% after 100 cycles. The main oxidation peak located at −0.22 V stays at the same position for the 100 cycles, while the main reduction peak located near −0.65 V continuously shifts toward the right and locates at −0.59 V at the end of the 100th cycle. For the N−450, the integral area of CV curves gradually decrease, which is maintained at 61% after 100 cycles. The main oxidation peak at −0.13 V shifts gradually toward −0.22 V at the first 10 cycles, and the peak position keeps steady at −0.22 V for the next 90 cycles. The main reduction peak located at −0.63 V continuously shifts toward −0.59 V, which is similar to the N−400. The displacement of redox peaks can be attributed to some irreversible phase transitions during cycling. Overall, the electrochemical capacity and cycling performance of the N−400 is better than the N−450, which is ascribed to the morphology, crystalline, and phase transitions.

The fitted and normalized electrochemical impedance test is used to further evaluate the electrochemical of the as-prepared V_6_O_13_ film electrode. The Figure 8 shows an equivalent circuit model according to the simulation. The model commonly employed in LIB studies consists of serial resistance (R_s_), charge transfer (R_ct_), a Warburg diffusion element (W_o_), and capacitive element (CPE). Generally, the impedance spectra can be divided into a semicircle in the high–frequency range that is assigned to the charge–transfer resistance at the electrode–electrolyte interface and a straight line in the low-frequency range that implies the Li^+^ diffusion-controlled process in the solid electrode [40,41]. The results demonstrate that the R_ct_ value of the N−400 is smaller than the N−450, indicating that the electrochemical reaction occurs more easily in the N−400 film. The reason can also be attributed to the morphology and crystallinity. Meanwhile, it is obvious that the R_ct_ value of both films decreases after cycling, which may be relevant to the gradual activation of the film electrodes that can expose more active sites after cycling, and phase transitions that occur during cycling.

To verify the morphology of the prepared films after 100 cycles, the SEM images are shown in Figure 9. The edges of the nanosheets of the N−400 have changed from flatter petal nanosheets to irregular nanosheets with cut edges, and the surface is rougher and some nanoparticles have grown on the surface. Figure 9c,d shows that the edges of the nanosheets have changed from smooth petal-shaped to rectangular stacked nanosheets with a slightly rougher surface than before. In summary, the overall morphology of the films was basically unchanged, but there was also a little self-aggregation and crushing, which means that the films were cyclically stable.

To verify the crystal phase structure and conjectures of the N-400 film during cycles, the ex-situ XRD patterns at different charge/discharge states are shown in Figure 10. Comparing them with the N-400 film before cycling, there is an obvious difference. After the first discharge process (Li^+^ insertion), the peak of V_6_O_13_ at 15.1° disappears and the peak intensity at 25.3° becomes weak, and new peaks at 13.1° (Li_0.6_V_1.67_O_3.67_, JCPDS No. 50-0230) and 22.8° (Li_3_VO_4_, JCPDS No. 39-0378) appear, indicating a phase transition during insertion of Li^+^. After the first discharge process, the peak intensity at 25.3° becomes stronger than the first charge process, which can be attributed to the extraction of Li^+^. The new phases of Li_0.6_V_1.67_O_3.67_ and Li_3_VO_4_ still exist, showing that the phase transition is not fully reversible. However, there are new peaks that appear at 27.6°, 36.5°, 44.7°, and 55.1° that can be indexed to the VO_2_ (JCPDS No. 82-0661) after the hundredth discharge process. Furthermore, the peak intensity of Li_0.6_V_1.67_O_3.67_, Li_3_VO_4_ becomes stronger and the peak at 25.3° of V_6_O_13_ basically disappears again, which is due to the insertion of Li^+^. Then, after the hundredth charge process (Li^+^ extraction), the peak intensity of Li_0.6_V_1.67_O_3.67_, Li_3_VO_4_, and VO_2_ decreases and the peak at 25.3° of V_6_O_13_ appears again.

The ex-situ XRD results indicate there are phase transitions during Li^+^ insertion and extraction. The process evolution mechanism is speculated as follows: Firstly, the phase starts to change when Li^+^ are inserted after the first discharge, the new phases of Li_0.6_V_1.67_O_3.67_ and Li_3_VO_4_ appear. That process is accompanied by a phase transition from V_6_O_13_ to VO_2_, the degree of this phase transition is very small, so the transformation of the crystal phase structure has not yet been achieved. After the first charge process, the transformation from V_6_O_13_ to VO_2_ still has not yet been completed. The Li_0.6_V_1.67_O_3.67_ and Li_3_VO_4_ still exist but the peak intensity becomes weak because the phase transition is not fully reversible with the extraction of Li^+^. Then after the hundredth discharge process, the transition from V_6_O_13_ to VO_2_ has been completed, so the peak of V_6_O_13_ is barely detected. When Li^+^ are extracted again after the hundredth charge process, the VO_2_ partly changes backward to V_6_O_13_ because of incomplete reversibility. Additionally, other phases of Li_0.6_V_1.67_O_3.67_ and Li_3_VO_4_ exhibit regular changes with Li^+^ insertion/extraction, and the peak intensity increases with ion insertion and decreases with ion extraction.

Figure 11 shows the Raman spectra after 100 cycles. The Raman peaks of the N−400 and N−450 are basically same, but there is an obvious difference with the film before cycling. A red shift of these Raman peaks both in N−400 and N−450 is observed after 100 cycles. The shift is attributed to the Li^+^ between the V-O layers, which results in a lattice expansion, and the split peak at 139 cm^−1^ may also related to it [30].

In order to obtain valence information for V, O, and Li of the films, XPS was carried out on the N−400 film before cycling and after 100 CV cycles. Figure 12a reveals that the N−400 film at different states both possess peaks of V and O, proving their coexistence directly. The obvious peak appearing at about 450 eV is the In of ITO substrate, which is due to the etching. Figure 12b shows the O 1s spectra of the N−400 film in different states. The O 1s spectra shift to the right after cycling, which can be attributed to the following two reasons: (1) The Li^+^ between the V-O layers bonding with O; (2) changes in the oxygen environment caused by etching onto the ITO glass substrate. Figure 12c shows that Li^+^ exist in both charge/discharge states, and the Li 1s is located at about 56 eV, which is the same as in other literatures [42,43]. The valence changes of V are shown in Figure 12d–f. It is obviously that the ratio of V^5+^/V^4+^ changes in different states. The N−400 film before cycling delivers a V^4+^/V^5+^ ratio of 1.9, which is basically consistent with V^4+^_4_V^5+^_2_O_13_, and is corresponding to the average valence of V (+4.33) in V_6_O_13_. After the hundredth discharge process, the V^4+^/V^5+^ ratio turns down to 1.3, because of the formation of the new Li_3_VO_8_ phase during cycling, which leads to the increase of the valence state of V. After the hundredth charge process, the V^4+^/V^5+^ ratio turns back to 1.4, which is caused by a part of the new Li_3_VO_8_ phase reversing back to V_6_O_13_. The results of the ex-situ XPS further prove the findings of the ex-situ XRD results and phase transitions during cycling.

## 4. Conclusions

In summary, nanosheet-like (NH_4_)_8_(V_19_O_41_(OH)_9_)(H_2_O)_11_ precursor films were synthesized via a simple low temperature liquid phase deposition method, and the V_6_O_13_ nanosheets film was successfully obtained by the following heat treatment under N_2_ atmosphere. The V_6_O_13_ films were directly used as cathodes for LIBs without adding binders and conductive agents. The results showed that the film annealed at 400 °C had the best electrochemical performance, with diffusion coefficients of 6.084 × 10^−12^ cm^2^s^−1^ (Li^+^ insertion) and 5.464 × 10^−12^ cm^2^s^−1^ (Li^+^ extraction), and with excellent cycling performance during CV cycles, which remained at 82.7% of capacity after 100 cycles. The ex-situ XRD results revealed the mechanism. There existed phase transitions with the insertion/extraction of Li^+^. The V_6_O_13_ partly transformed to Li_0.6_V_1.67_O_3.67_, Li_3_VO_4_, and VO_2_ with the insertion of Li^+^ into the lattice, and Li_0.6_V_1.67_O_3.67_, Li_3_VO_4_, and VO_2_ partly reversibly transformed backwards to V_6_O_13_ with the extraction of Li^+^. The phase transition was ascribed to the structure and nanosheet morphology of V_6_O_13_, which provided more ion diffusion paths and sufficient active sites during cycling. Moreover, there were enough space and diffusion channels in the structure for the phase transition. This work may provide an inspiration for enhancing the performance and studying the process mechanisms of vanadium oxides as cathode materials for LIBs.

## Figures and Tables

**Figure 1 materials-15-08574-f001:**
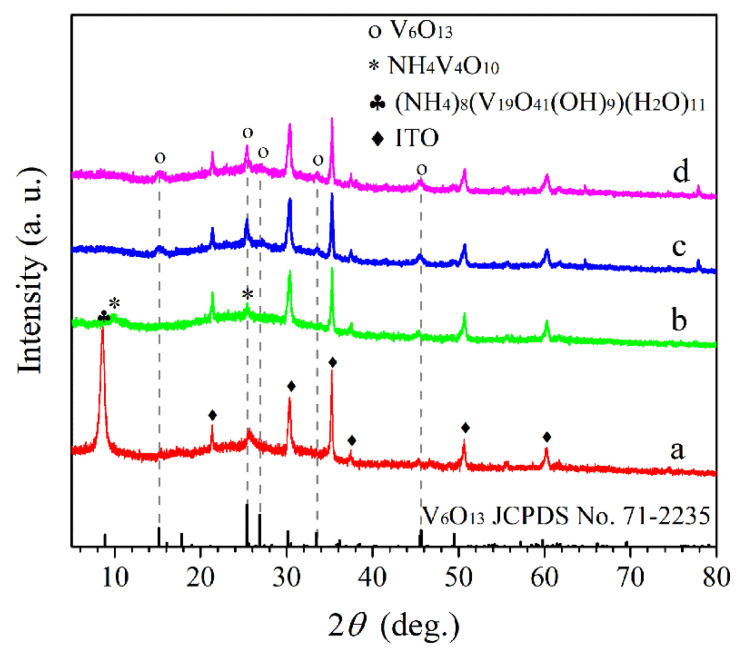
XRD patterns of films: the deposited film (a); N−350 (b); N−400 (c); N−450 (d).

**Figure 2 materials-15-08574-f002:**
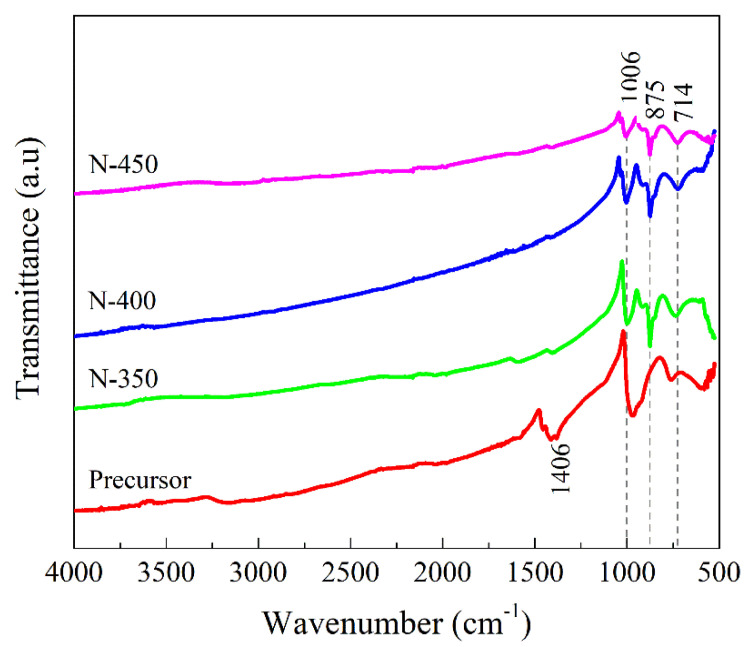
IR spectra of films N−350, N−400, and N−450.

**Figure 3 materials-15-08574-f003:**
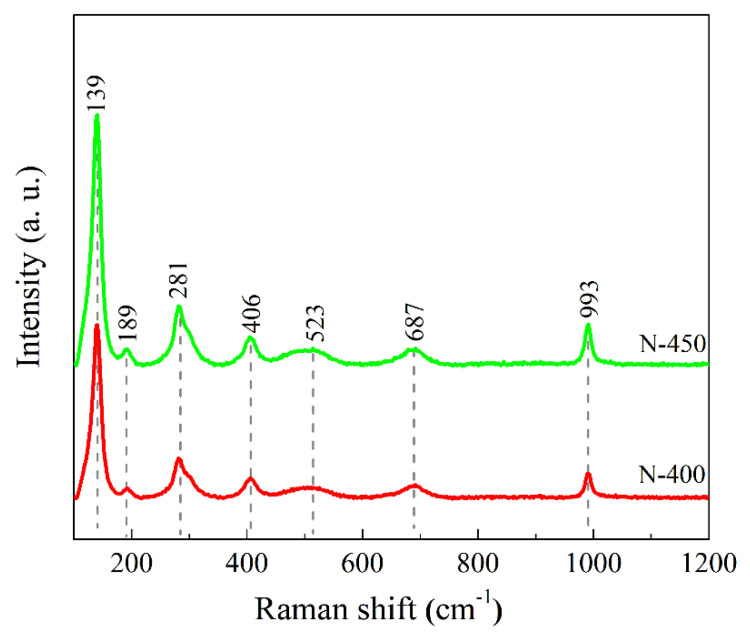
Raman spectra of films N−400 and N−450.

**Figure 4 materials-15-08574-f004:**
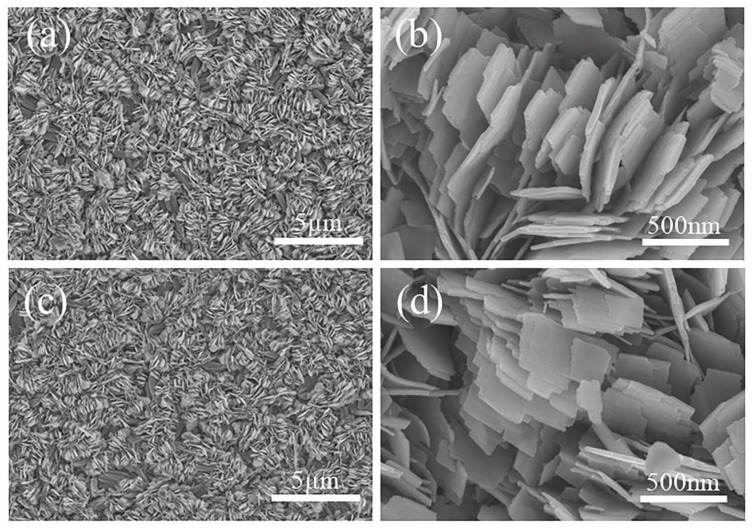
SEM images of the N−400 (**a**,**b**); N−450 (**c**,**d**).

**Figure 5 materials-15-08574-f005:**
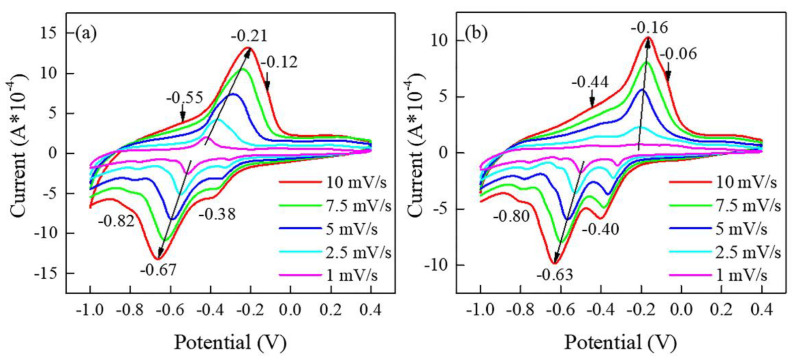
CV curves of the film-working electrodes at different scan rates: N−400 (**a**); N−450 (**b**).

**Figure 6 materials-15-08574-f006:**
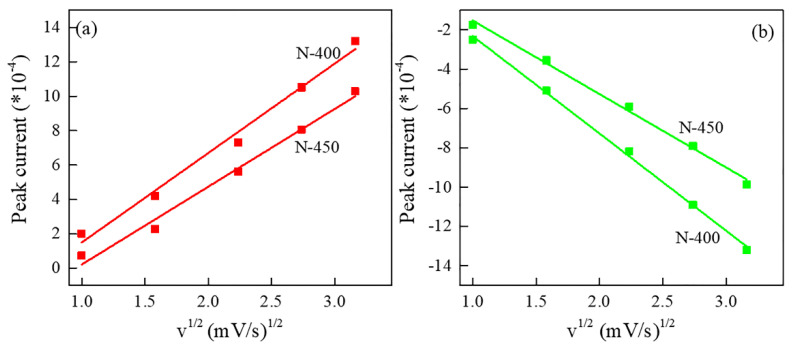
The linear fitting of the *I_p_* versus υ^1/2^ relationships for the anode (**a**) and cathode (**b**) electrodes.

**Figure 7 materials-15-08574-f007:**
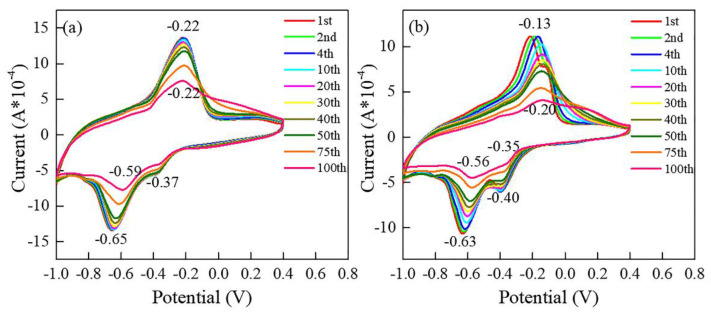
Cycling behaviors of the film-working electrodes at scan rate of 10 mV s^−1^: N−400 (**a**); N−450 (**b**).

**Figure 8 materials-15-08574-f008:**
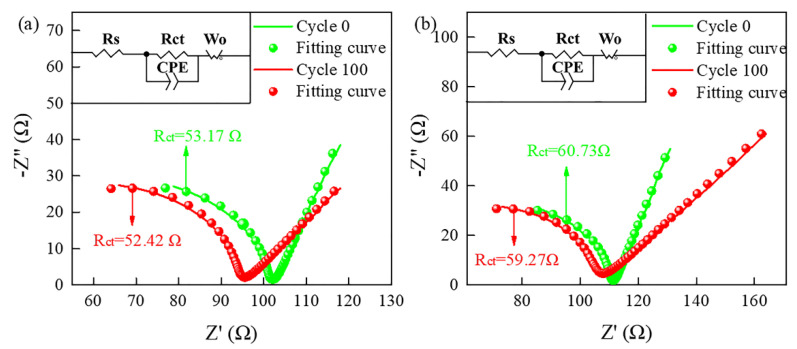
EIS diagram of films N−400 (**a**) and N−450 (**b**).

**Figure 9 materials-15-08574-f009:**
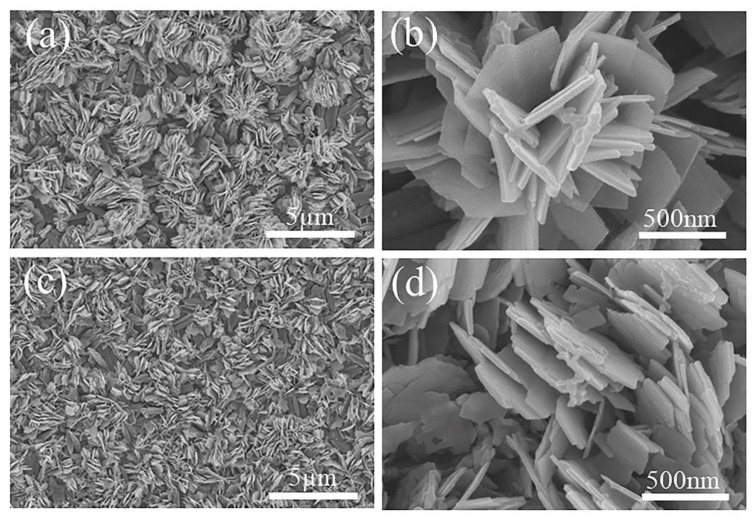
SEM images after 100 CV cycles of the films: N−400 (**a**,**b**); N−450 (**c**,**d**).

**Figure 10 materials-15-08574-f010:**
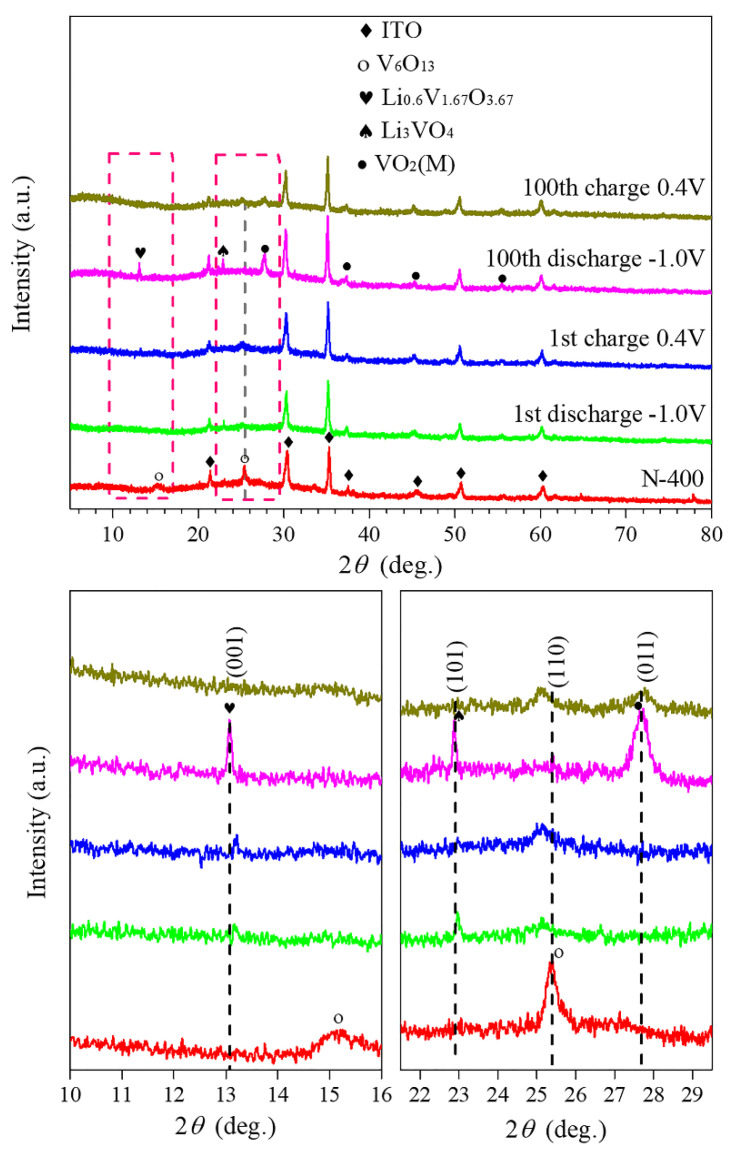
Ex-situ XRD patterns of the N−400 film.

**Figure 11 materials-15-08574-f011:**
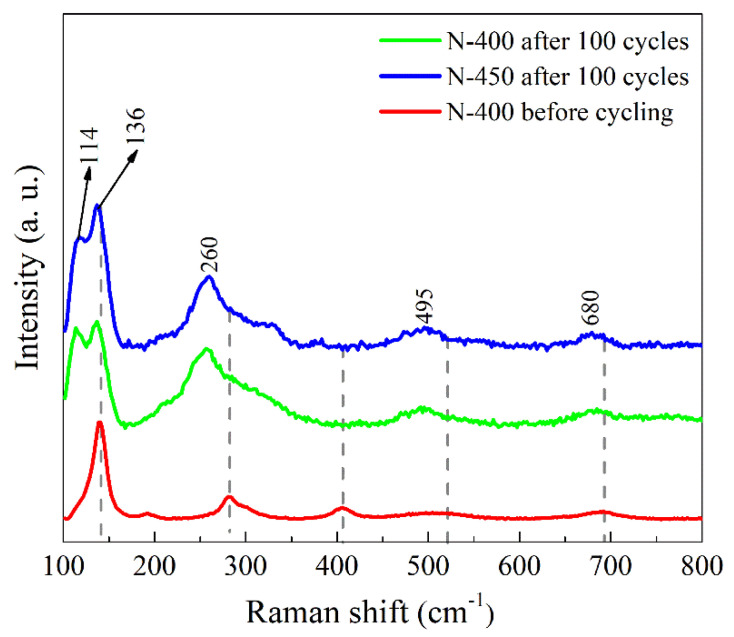
Raman spectra of the films after 100 cycles.

**Figure 12 materials-15-08574-f012:**
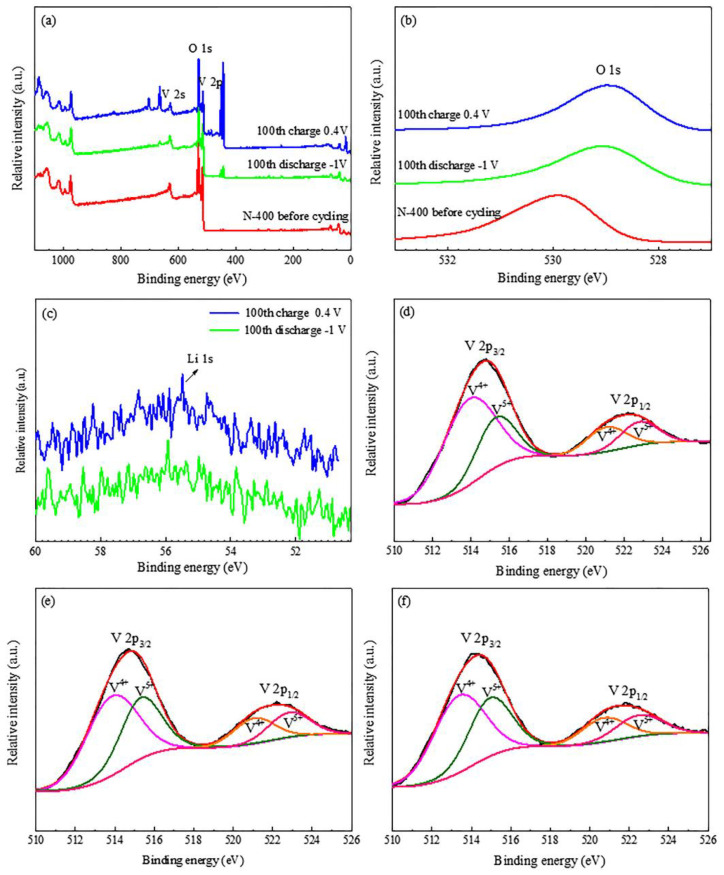
XPS spectra of the N−400 film: XPS spectrum before and after 100 cycles (**a**); O 1s before and after 100 cycles (**b**); Li 1s before and after 100 cycles (**c**); V 2p before cycling (**d**); after 100th discharge state (**e**); after 100th charge state (**f**).

**Table 1 materials-15-08574-t001:** Li^+^ diffusion coefficients at different annealing temperatures.

Anodic/Cathodic	Films	D_Li_^+^ (×10^−12^ cm^−2^s^−1^)
Anodic	N−400	5.65
	N−450	3.32
Cathodic	N−400	6.08
	N−450	3.20

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
