# Peer review of "Synthesis and Electrochemical Performance of V6O13 Nanosheets Film Cathodes for LIBs"

_materials, 2022, doi:10.3390/ma15238574_

Round 1

Reviewer 1 Report

The paper submitted by Fei Li. et. al describes the “Synthesis and Electrochemical Performance of V6O13 nanosheets film electrodes as cathodes for LIBs”. This paper is quite interesting from a technological point of view. I recommended Major revision of the manuscript.

Major suggestions below:

  1. The author synthesized the V6O13 nanosheets film with different temperature. What about high temperature for fabrication of V6O13 nanosheets film (above 500-600 °C)?
  2. The language is too poor, and the author should improve the quality of manuscript.
  3. The introduction is too short. Author should provide more information in the introduction part.
  4. The author should provide the high scan rates of CV for both samples.
  5. The author should provide the GCD curves of both samples.
  6. The author should prove the stability of GCD.

Author Response

  1. The author synthesized the V6O13nanosheets film with different temperature. What about high temperature for fabrication of V6O13nanosheets film (above 500-600 °C)?

Response: Thank you for your advice. The V6O15 nanosheets film was grown in-situ on ITO conductive glass. However, the ITO substrate is not able to withstand the high temperature, so we have chosen a calcination temperature of 450 ℃ and below as an insurance policy. Furthermore, the performance of N-450 has already turned worse than N-400. Therefore, the exploring experiments above 450 ℃ was not considered.

  1. The language is too poor, and the author should improve the quality of manuscript.

Response: We are sorry for the problems of language. We have tried to improve the language as much as possible to avoid confusion and disruption when you are reading.

  1. The introduction is too short. Author should provide more information in the introduction part.

Response: Thank you for your suggestions to make the article more complete. We have added some content to the introduction (page 4 and page 5).

4.The author should provide the high scan rates of CV for both samples.

Response: Thanks. The reason why we did not make CV tests at high scan rate is that the internal reaction of the electrode material is not complete. Therefore, at a lower scan rate, the polarization phenomenon can be reduced, the reaction can be more completed, and the oxidation reduction peak can be obviously observed.

  1. The author should provide the GCD curves of both samples.

Response: Thank you for your suggestion. The V6O13 film grown on ITO conductive glass is so thin that cannot be weighed for mass, we cannot use the specific capacity to express it. Therefore, we concentrate more on the phase change mechanisms during the CV cycles.

  1. The author should prove the stability of GCD.

Response: Thank you for your suggestion. We did not choose to do the GCD test because the film mass could not be weighed. The CV test for 100 cycles can also give the information of stability. The integral area (of the CV can represent the capacity to a certain extent, the overlap of the curves can represent the capacity fading, pointing to the stability.

Reviewer 2 Report

Overall mauscript is well written however few corrections are required for improving the quality of the journal. 

Line no: 53,55,57,58 for eg:  Zhong etc, specify the researchers is it etc  or et al clarify the sentence.

Line no: 119 in the figure 1b describes the peak at 9.2 is not clearly shown in the figure. 

Author Response

  1. Line no: 53,55,57,58 for eg:  Zhong etc, specify the researchers is it etc. or et al clarify the sentence.

Response: Thanks for your kindly reminder. We have corrected them in page 4 (Line 53, 55, 57 and 58).

  1. Line no: 119 in the figure 1b describes the peak at 9.2 is not clearly shown in the figure. 

Response: Thanks for your suggestion. We have carefully examined the Figure 1b. The crystal phase in Figure 1b is NH4V4O10, which is transformed from (NH4)8(V19O41(OH)9)(H2O)11 with temperature creases. Therefore, the crystallinity is not very complete and not very stable, which causes the peak at 9.2° is not strong.

Reviewer 3 Report

​Manuscript ID: materials-2015759

The manuscript entitled “Synthesis and Electrochemical Performance of V6O13 nanosheets film electrodes as cathodes for LIBs” by Li et al. has described the synthesis of V6O13 nanosheets film through a liquid phase deposition method on ITO glass substrates and utilization of the nanosheets as electrode material for LIBs. The authors need to address the following points to improve the MS,

1.      The XRD peaks are suggested to refine to understand the metal phases clearly since the ITO phases are predominant. Since V6O13 has been found to be pure phase, the crystal data and the crystal structure should be provided.

2.      The oxidation states should further be confirmed through XPS analyses.

3.      The surface area calculations should be provided properly.

4.      The charge-discharge study should be provided.

5.      How did the capacitive contributions of the electrode materials vary with scan rate?

6.      The redox reactions that are going on at the electrode, should be discussed properly. The discussion section should be rewritten with proper explanations.

7.      The stability and effectiveness studies should be provided.

8.      SEM images of materials before and after 100 cycles should be provided at the same magnifications.

Author Response

  1. The XRD peaks are suggested to refine to understand the metal phases clearly since the ITO phases are predominant. Since V6O13has been found to be pure phase, the crystal data and the crystal structure should be provided.

Response: We appreciate your suggestion. The crystal data and the crystal structure have been provided in page 7. The monoclinic V6O13 phase (JCPDS No. 71-2235) with the lattice parameters of a=11.922 Å, b=3.68 Å, c=10.138 Å, and β=100.87° (page 7).

  1. The oxidation states should further be confirmed through XPS analyses.

Response: Thanks for your reminder. We have further analyzed the XPS results, including the peak splitting fits for valence state of vanadium, and phase changes corresponding to redox of vanadium.

  1. The surface area calculations should be provided properly.

Response: Thanks for your suggestion. We have refitted the results of XPS and added surface area calculations of at different charge/discharge state (page 21). According to the combination valence state of vanadium and the phase transition during cycling, the previous analysis and discovery were further determined.

  1. The charge-discharge study should be provided.

Response: Thank you for your valuable suggestion. The mass of the film is difficult to weigh and its mass after removing the glass substrate is in the fourth position of the scale, sometimes even less than before. Therefore, we cannot use the specific capacity to express it. The charge/discharge test is not covered here, we focus more on the mechanistic changes during the CV test, which is also studied later.

  1. How did the capacitive contributions of the electrode materials vary with scan rate?

Response: Thank you for your comments. The scan rate can affect the speed of diffusion of the ions, which results in a varying number of ions insertion/extraction. Therefore, the capacitive contributions of the electrode materials vary with scan rate. The specific explanations are as follows: The capacity contributions of the electrode materials vary with scan rate can be obtained according to the following equation:

In this equation, i is peak current, v is scan rate. The plot of log(i) varies linearly with log(v), the slope of the fitting line equals to the value of b. The b value of 0.5 indicates that the current is totally attributed to diffusion, while b =1 reveals a complete capacitive behavior. When b value between 0.5 to 1, indicating that current is influenced by both capacitance and ions diffusion. The percentage of the impact of the two can be calculated according to the following formula:

The capacitive contribution is k1, and diffusion contribution is k2. Therefore, changes in scan rate will lead to changes in capacity.

  1. The redox reactions that are going on at the electrode, should be discussed properly. The discussion section should be rewritten with proper explanations.

Response: Thanks. We have already discussed properly about the redox reactions that going on at the electrode. We have re-organized the thinking and explained the reactions that occur during redox reactions, hoping to express our ideas more clearly (page 11 and page 14).

  1. The stability and effectiveness studies should be provided.

Response: Thanks for your suggestion. The mass of the film is difficult to weigh. Therefore, we cannot use the specific capacity to express it. But the CV test for 100 cycles can also give some indication of stability. The integral area of the CV can represent the capacity to a certain extent, the information on the stability of the films can also be obtained from some available data. In CV test for 100 cycles (page 14), the overlap of the curves can represent the decay of the capacity, pointing to the stability.

  1. SEM images of materials before and after 100 cycles should be provided at the same magnifications.

Response: Thanks for your reminder. For a clearer comparison, we have provided the SEM images at the same magnifications before (page 10) and after 100 cycles (page 16).

Round 2

Reviewer 1 Report

The manuscript can be accepted in current format.

Author Response

Thank you again for your comments concerning our manuscript entitled “Synthesis and Electrochemical Performance of V6O13 nanosheets film electrodes as cathodes for LIBs”. We have carefully checked and corrected based on the comments. The revised parts are marked in red in the manuscript. The responses to the comments and the revisions are listed below.

Reviewer 1: English language and style are fine/minor spell check required.

Response: We have carefully checked and improved the language and style. Thanks for your kindly reminder.

Once again, thank you very much for your comments and suggestions.

Sincerely yours,

Haiyan Xu

Reviewer 3 Report

The MS has been improved along with explanations however, the elctrochemical surface area of the materials should be reported.

Author Response

Thank you again for your comments concerning our manuscript entitled “Synthesis and Electrochemical Performance of V6O13 nanosheets film electrodes as cathodes for LIBs”. We have carefully checked and corrected based on the comments. The revised parts are marked in red in the manuscript. The responses to the comments and the revisions are listed below.

Reviewer 3: The MS has been improved along with explanations however, the electrochemical surface area of the materials should be reported.

Response: Thanks for your suggestion. The electrochemical surface area in this work is 0.6 cm2, which is mentioned in the experimental section (page 6).

Once again, thank you very much for your comments and suggestions.

Sincerely yours,

Haiyan Xu
